# Genome and Secretome Analysis of *Staphylotrichum longicolleum* DSM105789 Cultured on Agro-Residual and Chitinous Biomass [note 1]

**DOI:** 10.3390/microorganisms9081581

**Published:** 2021-07-25

**Authors:** Arslan Ali, Bernhard Ellinger, Sophie C. Brandt, Christian Betzel, Martin Rühl, Carsten Wrenger, Hartmut Schlüter, Wilhelm Schäfer, Hévila Brognaro, Martin Gand

**Affiliations:** 1Institute of Biochemistry and Molecular Biology, University of Hamburg, Martin Luther King Platz 6, 20146 Hamburg, Germany; arslanali1986@gmail.com (A.A.); Christian.Betzel@uni-hamburg.de (C.B.); cwrenger@icb.usp.br (C.W.); hschluet@uke.de (H.S.); hevila.brognaro@chemie.uni-hamburg.de (H.B.); 2Dr. Panjwani Center for Molecular Medicine and Drug Research, International Center for Chemical and Biological Sciences, University of Karachi, University Road, Karachi 75270, Pakistan; 3Institute of Clinical Chemistry and Laboratory Medicine, Diagnostic Center, Section Mass Spectrometry & Proteomics, Campus Research, Martinistr. 2, N27, Medical Center Hamburg-Eppendorf, Universität Hamburg, 20246 Hamburg, Germany; 4Department ScreeningPort, Fraunhofer Institute for Translational Medicine and Pharmacology ITMP, Schnackenburgallee 114, 22525 Hamburg, Germany; Bernhard.Ellinger@itmp.fraunhofer.de; 5Department of Molecular Phytopathology, Biocenter Klein Flottbek, University of Hamburg, Ohnhorststr. 18, 22609 Hamburg, Germany; sophie.brandt@live.de (S.C.B.); wilhelm.schaefer@gmail.com (W.S.); 6Institute of Food Chemistry and Food Biotechnology, Department Biology and Chemistry, Justus Liebig University Giessen, Heinrich-Buff-Ring 17, 35392 Gießen, Germany; Martin.Ruehl@lcb.Chemie.uni-giessen.de; 7Biomedical Science Institute, University of São Paulo, Av. Lineu Prestes, 2415, São Paulo CEP 05508-900, Brazil

**Keywords:** *Staphylotrichum longicolleum*, genome analysis, secretome, mass spectrometry, CAZyme analysis, chitin degradation, residual biomass treatment

## Abstract

*Staphylotrichum longicolleum* FW57 (DSM105789) is a prolific chitinolytic fungus isolated from wood, with a chitinase activity of 0.11 ± 0.01 U/mg. We selected this strain for genome sequencing and annotation, and compiled its growth characteristics on four different chitinous substrates as well as two agro-industrial waste products. We found that the enzymatic mixture secreted by FW57 was not only able to digest pre-treated sugarcane bagasse, but also untreated sugarcane bagasse and maize leaves. The efficiency was comparable to a commercial enzymatic cocktail, highlighting the potential of the *S. longicolleum* enzyme mixture as an alternative pretreatment method. To further characterize the enzymes, which efficiently digested polymers such as cellulose, hemicellulose, pectin, starch, and lignin, we performed in-depth mass spectrometry-based secretome analysis using tryptic peptides from in-gel and in-solution digestions. Depending on the growth conditions, we were able to detect from 442 to 1092 proteins, which were annotated to identify from 134 to 224 putative carbohydrate-active enzymes (CAZymes) in five different families: glycoside hydrolases, auxiliary activities, carbohydrate esterases, polysaccharide lyases, glycosyl transferases, and proteins containing a carbohydrate-binding module, as well as combinations thereof. The FW57 enzyme mixture could be used to replace commercial enzyme cocktails for the digestion of agro-residual substrates.

## 1. Introduction

The sustainable valorization of non-edible lignocellulosic biomass facilitates the production of fuels, chemicals, and other carbon-based materials while avoiding competition with food and feed crops [1,2]. Abundant biomass can be obtained from forestry and agricultural waste, such as sugarcane in tropical areas and maize in sub-tropical and temperate regions [3,4,5]. Side streams and waste from the fishing industry can also be used to reduce waste and unlock new biobased resources [6].

Lignocellulosic biomass is heterogeneous in structure and composition, depending on the plant species [7,8,9]. This can impede the enzymatic hydrolysis of biomass [10,11]. Cellulose, consisting of linear chains of β-(1,4)-d-glucose, is the most abundant polymer on earth and the main component of lignocellulosic biomass. The second most abundant component is hemicellulose, comprising at least six different macromolecules [12,13]. These are: (a) xylans with a linear backbone of β-(1,4)-linked β-d-xylopyranosyl residues, (b) glucuronoxylans consisting mainly of 4-*O*-methyl-α-d-glucuronopyranosyl units, (c) arabinoxylans containing xylose substituted with α-l-arabinofuranosyl units, (d) xyloglucans containing a cellulose-like linear backbone with additional β-(1,6)-linked xylose sidechains, often terminally fused with other sugars such as galactose and fucose [14]; (e) glucomannans [15] and (f) galactoglucomannans [16], each featuring backbones of β-(1,4)-linked d-mannose and d-glucose, the latter with additional α-(1,6)-linked galactose units and both sometimes branched with β-(1,6)-glucosyl residues. Pectin, the third most abundant component of lignocellulosic biomass, is a complex heteropolymer divided into three subclasses consisting of 12 types of glycosyl units that form at least 22 types of glycosidic bonds [17]. Three examples highlight the extraordinary diversity of pectin structures: (a) homogalacturonan comprises linear α-(1,4)-d-galacturonic acid chains partly esterified with methyl groups; (b) rhamnogalacturonan-I with repeated disaccharides of galacturonic acid and rhamnosyl residues, and linear or branched α-l-arabinofuranosyl and/or galactopyranosyl side chains on C-4; and (c) rhamnogalacturonan-II, which is similar but the galacturonans feature different side-chain residues.

Due to the complexity of lignocellulosic biomass, enzymatic hydrolysis requires efficient palettes of enzymes that can break down cellulose as well as hemicellulose [18] and pectin [19], but the enzymes are hindered by the inaccessibility of the substrates. This can be addressed by physical and/or chemical pretreatment, but these processes generate toxic molecules and inhibitors that limit enzymatic activity, therefore interfering with subsequent fermentation processes [20]. Enzymatic pretreatment can serve as an alternative [21,22,23], but in contrast to physical and/or chemical pretreatment it requires adaptation to the type of biomass. Even if the polysaccharide content of the cell walls is similar, the cross-linking and interactions between polysaccharide and lignin/phenolic compounds can still vary depending on the plant species [24,25]. This is the case for maize leaves and sugarcane culm, which have a similar polysaccharide content but differ in terms of cell wall thickness, cellulose crystallinity, and the content of hemicellulose, pectin and lignin [26].

Chitin, a by-product of the shellfish industry, is the second most abundant biopolymer after cellulose, with more than 10^11^ tons produced naturally per year [27]. Chitin is also the main constituent of the exoskeletons of insects and mollusks, and fungal cell walls [28]. Like cellulose, chitin is a linear polymer with β-(1,4) linkages, but the monomeric unit is *N*-acetyl-d-glucosamine rather than d-glucose [29]. There are three forms of chitin (α, β and γ), which differ in their degree of hydration, unit cell size, and the number of chitin chains per unit cell. The most abundant is α-chitin, which has a crystalline structure with anti-parallel sheets, whereas β-chitin consists of parallel sheets and γ-chitin is a combination. Chitosan is produced industrially by the deacetylation of chitin [30]. Chitinous biomass has many applications in agriculture and horticulture [31], as a source for advanced functional polymers [32], and as the basis of drug delivery systems and wound dressings [33]. The remaining waste biomass could be used as a substrate for biofuel production [34].

The breakdown of abundant natural polymers such as lignocellulose and chitin is one of the main ecological functions of fungi, making them promising candidates for the discovery of enzymes or enzyme consortia for biomass degradation [35]. More than 5 million species of fungi have been described thus far, and the number is likely to increase given that only 5% of species are formally classified [36,37]. The filamentous fungal strain we investigated in this study is an ascomycete in the class Sordariomycetes, order Sordariales, and family *Chaetomiaceae*, therefore representing the most general fungal subkingdom Dikarya, which includes the large phyla Ascomycota (ascomycetes, sac fungi) and Basidiomycota (basidiomycetes, higher mushrooms or pillar fungi) [38]. More than 450 species of *Chaetomiaceae* have been described [39], the first in 1817 [40]. *Chaetomiaceae* are soil-borne, saprotrophic, endophytic and pathogenic species that adapt rapidly to various growth conditions [41]. At least 18 different lineages are recognized, some of which produce thick-walled spores (*Humicola sensu stricto*, *Mycothermus*, *Staphylotrichum*, and *Trichocladium*) similar to the species described in this study. Three genomes of related strains (*Chaetomium thermophilum*, *Chaetomium globosum* and *Chaetomium cochliodes*) have fully sequenced genomes, ranging in size from 28.3 to 34.9 Mbp and containing 7165 to 11,048 predicted open reading frames (ORFs) [41,42,43].

Our previous analysis of 295 fungal isolates, collected from different substrates and various environments in Vietnam, revealed their ability to degrade lipids, chitin, cellulose and xylan [44]. Four isolates were able to digest chitin with remarkable efficiency, three of which were *Aspergillus* sp. strains, and the other was the less studied *Chaetomiaceae* strain FW57, originally isolated from dead mangrove wood. We therefore selected strain FW57 for robust analysis using a four-locus phylogeny, resulting in its assignment to the species *Staphylotrichum longicolleum*, formerly known as *Chaetomium longicolleum* [45]. We characterized FW57 in detail by genome and secretome analysis, leading to the identification of undiscovered lignocellulose and chitin degrading enzymes and other carbohydrate-active enzymes (CAZymes) with the ability to convert sugarcane bagasse and maize leaves into fermentable sugars.

## 2. Materials and Methods

### 2.1. Fungus Isolation and Growth Conditions

The fungal isolate *Staphylotrichum longicolleum* FW57 was obtained from mangrove wood [44] in Vietnam (longitude 10°36′015N, latitude 106°56′045E) and a conidial suspension was used for storage and downstream experiments. A mycelium piece (5 mm diameter) from potato dextrose agar (PDA) was transferred to a fresh PDA plate and grown in the dark for 5–7 days at 28 °C. The conidia were scraped with a Drigalski cell spreader and sterile water, and centrifuged at 2693× *g* for 20 min at 4 °C. The pellet was washed twice with sterile water, resuspended and filtered through a 40-µm mesh sieve. After repeating this process, the resuspended pellet was aliquoted and stored at −70 °C. To investigate mycelial growth and possible color formation, fungal growth was assessed on PDA, yeast extract peptone dextrose (YPD) [46], complete medium (CM) [47], malt extract agar (MEA) [48], starch casein agar (SCA) [49] and Mandels’ salt medium (MS) [50], each with 2% agar, for 15 days (Figure 1).

### 2.2. Phylogenetic Analysis and De Novo Sequencing

Submerged cultures of *S. longicolleum* FW57 were established in potato dextrose broth (PDB) and incubated at 28 °C, shaking at 150 rpm. DNA was isolated using the CTAB method [51,52] and its purity and quality were confirmed by gel electrophoresis and spectrophotometry. We used 11 μg of pure high-molecular-weight genomic DNA for the de novo preparation of 270-bp short HiSeq and PACBIO RSII 20K sequencing libraries (Beijing Genomics Institute, China). Gene prediction, genome assembly, and annotation were carried out using the funannotate software package v1.7.4 (https://doi.org/10.5281/zenodo.3679386, accessed on 10 October 2020), which included the assignment of BUSCO groups (Benchmarking Universal Single-Copy Orthologs), Pfam domains, proteases, and CAZymes. The ITS-1/8S rRNA/ITS-2 region was amplified and sequenced using primers ITS1_fw (5′-TCC GTA GGT GAA CCT GCG G-3′) and ITS4_rv (5′-TCC TCC GCT TAT TGA TA TGC-3′) as previously described [53] and the ITS sequence was deposited in GenBank (accession number MG098702). The four multiple sequence alignments for marker genes *RPB2*, *TUB2*, *ITS* and *D1D2_LSU* [45] across 173 taxa and 864, 1125, 701 and 574 alignment columns were kindly provided by Jos Hoebraken (personal communication). We built four independent covariance models using cmbuild v1.1.3 in the Infernal package (https://doi.org/10.1093/bioinformatics/btt509, accessed on 11 November 2020) from the sequence alignments without consensus structure information (parameter -noss). The bit scores depend on multiple sequence alignment length (more precisely, the covariance model length), so we ran the ungapped alignment sequences against their covariance models (cmalign -noss -g) and obtained 769, 488, 495, and 596 bits as average scores for *RPB2*, *TUB2*, *ITS* and *D1D2_LSU*, respectively. Given that a covariance model without a consensus structure is basically a hidden Markov model (HMM), we initially used hmmbuild and hmmsearch (www.hmmer.org, accessed on 9 December 2020) instead, but this did not yield hits with sufficient scores, most likely due to high penalties for the insertion of introns.

Using covariance models on the FW57 hybrid Pacbio+Illumina assembly, we recovered good hits for *RPB2* on Scaffold 4 at position 3,523,318–3,524,169 (813 bits); for *TUB2* also on Scaffold 4 at position 1,958,535–1,959,189 (488 bits); for *ITS* on Scaffold 1 at position 12,160,562–12,161,080 (520 bits); and for *D1D2_LSU* on Scaffold 1 at position 12,161,081–12,161,641 (595 bits).

We next aligned the four identified marker gene regions to the initial sequence alignment (Appendix A) and used IQTree v1.6.12 (https://doi.org/10.1093/molbev/msaa015, accessed on 11 November 2020) with the settings -nt AUTO -bb 5000, and partitions *RPB2* = 1–864, *TUB2* = 865–1989, *ITS* = 1990–2690 and *D1D2_LSU* = 2691–3264 to construct the phylogenetic tree with partitioned maximum likelihood bootstrapping. The resulting Newick tree file (Appendix A) was rooted at *Microascus trigonosporus* strain CBS 218.31 [45] and colored using FigTree v1.4.4 (http://tree.bio.ed.ac.uk/software/figtree/, accessed on 11 November 2020) as shown in Figure 2.

### 2.3. Function Annotation and CAZyme Analysis

To predict protein functions, all 31,910 coding sequences (CDS) from our genome assembly were annotated in Pfam v33.1 using HMMER to predict the closest protein domains and transfer their functional annotations. We also used 10,990 CDS as homology search queries against CAZyme database v53 in the standalone version of the dbCAN annotation tool (run_dbCAN.py [54]), which internally used DIAMOND v2.0.6, Hotpep (version from 2018-04-23) and HMMER v3.3.1. To increase sensitivity, we manually performed additional BLASTP (v2.9.0+) searches for CDS marked as homologs by only one of the three programs in run_dbCAN. We considered all CDS as true hits if they were predicted by at least two tools (Appendix A). CAZyme families often contain members with diverse EC classifications, so we applied two strategies to ascertain the functional activity: first, we scored all the above hits against Pfam to predict the closest protein domains and transfer their functional annotations; and second, the hits were individually used as BLASTP (v2.9.0+) queries against the database of all CAZyme sequences, and EC annotations were taken from the best scoring hit. This did not resolve ambiguity in all cases, but offered reasonable functional predictions (Appendix A).

### 2.4. Secretome Analysis and SDS-PAGE

The *S. longicolleum* secretomes were induced by two different types of fermentation. For liquid fermentation, we used 100-mL Erlenmeyer flasks and the mycelia were pre-cultivated in YPD medium at 28 °C for 3 days, shaking at 150 rpm. Afterwards, mycelia were washed briefly and dried between sheets of filter paper (Whatman, Dassel, Germany). We then incubated 0.1 g of the semi-dried mycelia with 50 mL inductive medium at 28 °C for 72 h, shaking at 150 rpm (in triplicate). The inductive medium was composed of mineral salts (0.35% NaNO_2_, 0.15% K_2_HPO_4_, 0.05% MgSO_4_ × 7H_2_O, 0.05% KCl, 0.001% FeSO_4_ × 7H_2_O) supplemented with 1% (*w*/*v*) chitinous biomass, namely chitin (C) or high-, medium-, and low-molecular-weight forms of chitosan (described hereafter as H-CS, M-CS and L-CS), all from Sigma-Aldrich (Steinheim, Germany). For solid-state fermentation (SSF), we used a combination of agro-residual biomass with Vogel medium [55]. Sugarcane bagasse fibers treated by steam explosion (XSCB) [56] and maize leaves (MZ) were washed, dried in an oven (60 °C for 24 h) and milled to a maximum diameter of 1.4 mm. We then placed 1 g of dry agro-residual biomass in 20 × 30 cm polypropylene bags (Carl Roth, Karlsruhe, Germany) attached to a plastic tube bottleneck, and sterilized the samples at 120 °C for 40 min. *S. longicolleum* FW57 mycelia were prepared by static fermentation for up to 12 days in autoclaved (120 °C for 20 min) medium containing 2.4% potato extract and 0.7% yeast extract. For inoculation, 200 mL of the medium was inoculated with 10 5-mm agar discs cut from 7-day-old plates prepared with the same yeast and potato extracts plus 2% agar. After 12 days, the thick layer of mycelia was washed three times using autoclaved ultrapure water, suspended and stirred in 100 mL sterile Vogel medium [55] and finally 15 mL of mycelial biomass was added to the 1 g of solid substrate (equivalent to 1.5 g dry mycelial biomass per gram of solid biomass). SSF was carried out at 37 °C for up to 28 days. Two bags representing each substrate were taken at 7, 14, 21, and 28 days. The secretome was obtained by mixing 10 mL of distilled water with the fermented solid substrate and stirring for 2 h at 4 °C, before filtration and centrifugation (3250× *g*, 30 min, 4 °C) to remove the fungal biomass. The resulting fungal supernatants were used for secretome analysis and enzymatic activity testing. Supernatants from all sources were separated by SDS-PAGE on 12% polyacrylamide gels [57] followed by staining with 0.1% Coomassie Blue R250 and destaining with 45% methanol and 10% acetic acid before proteomic analysis. Remaining samples were retained for the analysis of enzyme activities.

### 2.5. Proteomics

#### 2.5.1. Sample Preparation

In-gel tryptic digestion [58] was carried out by dividing each lane of the gel into 4–5 equal parts and dicing them, followed by reduction (10 mM dithiothreitol in 100 mM ammonium bicarbonate), alkylation (55 mM iodoacetamide in 100 mM ammonium bicarbonate) and digestion with 13 ng/µL trypsin (Promega, Mannheim, Germany) in 10 mM ammonium bicarbonate containing 10% (*v*/*v*) acetonitrile. Tryptic peptides were extracted with a 1:1 mixture of 5% formic acid and acetonitrile and were completely lyophilized. The peptides were resuspended in 40 µL 0.1% formic acid prior to LC-MS/MS analysis. For the time-course analysis of the fungal supernatants from SCB or MZ substrates at 7, 14, 21, and 28 days, proteomic samples were obtained by in-solution digestion. Approximately 20 µg of protein was mixed with 6 M urea in 100 mM NH_4_HCO_3_ followed by reduction (100 mM DTT) and alkylation (300 mM iodoacetamide) and digestion (0.25 μg/μL trypsin). Samples were cleaned using Sep-Pak C18 SPE cartridges (Waters, Milford, MA, USA). After lyophilization, the tryptic peptides were resuspended in 20 µL 0.1% formic acid prior to LC-MS/MS analysis.

#### 2.5.2. LC-MS/MS Analysis of the Secretomes

We injected 1–3 µL of the samples onto an Acclaim PepMap C-18 nanoViper trapping column (100 μm × 20 mm, 5 μm, 100 Å; Thermo Fisher Scientific, Waltham, MA, USA) at a flow rate of 3 μL/min and washed for 5 min with 98% buffer A (0.1% formic acid in MS-grade water) and 2% buffer B (0.1% formic acid in acetonitrile). The peptides were separated on an Acclaim PepMap C-18 nanoViper reversed-phase capillary column (75 µm × 25 cm, 2 µm, 100 Å; Thermo Fisher Scientific) at 45 °C using a Dionex Ultimate 3000 nano-UPLC system (Thermo Fisher Scientific) connected to an Orbitrap Fusion tribrid (quadrupole/Orbitrap/linear ion-trap) mass spectrometer or Waters nanoAcquity nano-UPLC system connected to a Q Exactive hybrid (quadrupole/Orbitrap) mass spectrometer (Thermo Fisher Scientific). The gradient system consisted of buffers A and B at a constant flow rate of 300 nL/min for 70 min. The profile was held at 3% B for 5 min followed by a gradient to 28% B at 35 min, then 35% B at 40 min, and 90% B at 40 min 6 s. After a hold at 90% B for 9 min 54 s, the column was equilibrated at 3% B for 19 min 54 s. Eluted peptides were ionized in positive ion mode using a nanospray Flex with an electrospray ionization source (Thermo Fisher Scientific) and a fused-silica nano-bore emitter with an internal diameter of 10 μm (New Objective, Woburn, MA, USA) at a capillary voltage of 1800 V for both mass spectrometers. For the Orbitrap Fusion mass spectrometry system, the ion transfer tube temperature was set to 300 °C. Parent ion scans were carried out in the range 400–1300 *m/z* in the Orbitrap mass analyzer at 120 K resolution with a maximum injection time of 120 ms and an automatic gain control (AGC) target value of 2 × 10^5^. Data-dependent acquisition (DDA) mode was set to top speed for precursor ion selection. The most intense peaks (intensity threshold = 5 × 10^3^) were isolated with a quadrupole isolation width of 1.6 *m/z*, fragmented by high-energy collisional dissociation (collision energy = 30%) and detected in the ion-trap mass analyzer. A dynamic exclusion filter was applied for 30 s and excluded after one time. For ion-trap detection, the scan rate was set to a rapid scan range of 400–1300 *m/z*. The maximum injection time was 60 ms, and the AGC target value was 1 × 10^4^. For the Q Exactive mass spectrometer, the ion transfer tube temperature was set to 275 °C and the full scan was acquired with a resolution of 70,000 full width at half maximum (FWHM) at 200 *m/z* (MS level) over a scan range of 400–1300 *m/z.* The maximum injection time was 120 ms and the AGC target value was 1 × 10^6^. For DDA, the most intense ions were isolated to an AGC target value of 5 × 10^5^ using Top 12 mode with a maximum injection time of 60 ms and a resolution of 17,500 FWHM at 100 *m/z*. Fragmentation was carried out with a high-energy collisional dissociation of 27% with precursor selection at an intensity threshold of 1x 10^5^ and a 2.0 *m/z* isolation window. Precursors were selected for fragmentation with charge states +2 and higher. Dynamic exclusion was applied for 30 s and excluded after one time.

#### 2.5.3. Protein Identification by Database Matching

The LC-MS/MS data files were screened against the database of *S. longicolleum* DSM105789 translated sequences (Appendix A) using Proteome Discoverer v2.0 (Thermo Fisher Scientific) including the search engine Sequest HT. The search parameters included precursor and product ion mass tolerances of 10 ppm and 0.5 Da (or 0.02 Da for Q Exactive data), respectively, two missed cleavages allowed, cysteine carbamidomethylation as a fixed modification, and methionine oxidation as a variable modification. Proteins found with at least one unique peptide and a false discovery rate (FDR) of 1% (determined by percolator) were accepted [58].

### 2.6. Enzymatic Activity

Enzymatic hydrolysis was measured using the dinitrosalicylic acid (DNS) method [59] for samples taken every 7 days after SSF at 45 °C with the following substrates: 0.1% wheat arabinoxylan (Megazyme, Bray, Ireland), 0.5% beechwood xylan, 0.5% carboxymethylcellulose, 1% polygalacturonic acid or 0.2% citrus pectin and laminarin (all from Sigma-Aldrich). We mixed 10 µL of the *S. longicolleum* extract with 50 µL of each substrate and 40 µL 50 mM citrate buffer (pH 4.8). Xylan was assayed for 10 min and the remaining substrates for 3 h. Filter paper activity (FPase) was determined as previously described [60] with modifications so that the scale of the reaction was reduced 10-fold for all reactants. Reducing sugars were measured by the DNS method, as described above, using glucose standards. Chromogenic substrates *p*-nitrophenyl-β-d-glucopyranoside and *p*-nitrophenyl-β-d-cellobioside (1 mM) were used to measure β-glucosidase and β-cellobiohydrolase activity as previously described [61]. Briefly, incubation was carried out at 45 °C for 10 min and the reaction was stopped by adding 100 μL 1 M Na_2_CO_3_. Absorbance was measured at 410 nm and the concentration of released *p*-nitrophenol was used to calculate the enzymatic activity using a *p*-nitrophenol standard curve.

Chitinase activity was determined following liquid fermentation using either mineral salt medium [47] or YPD medium (28 °C, shaking at 145 rpm in the dark) for 3 days or every 24 h for up to 5 days followed by centrifugation (3250× *g*, 4°, 20 min) to obtain the cultivation supernatant. Chitinase activity was measured in 1.5 mL of supernatant, which was incubated with 1.5 mL 2% (*w*/*v*) powdered chitin from shrimp shells (Sigma-Aldrich) in 50 mM sodium acetate (pH 6.5) at 37 °C for 2 h, shaking at 300 rpm. Released *N*-acetylglucosamine was quantified using Schales reagent [62]. Protein concentration was determined using the ROTI Nanoquant protein detection kit (Carl Roth) by adding 50 μL of the supernatant to 200 μL of the detection solution. Measurements were taken from at last three experimental replicates. For activity calculation, one unit (U) was defined as the amount of enzyme required to release one μmol of product per minute under the assay conditions.

### 2.7. Saccharification of Sugarcane Bagasse and Maize Leaves

The conversion of 5% (*w*/*v*) in-nature sugarcane bagasse (NSCB), steam-exploded sugarcane bagasse (XSCB) [56] and maize leaves (MZ) into glucose was assessed using the *S. longicolleum* FW57 secretome obtained after SSF for 21 days. For the in-house enzymatic mixture, the lyophilized secretome fractions from both biomass substrates were resuspended in 50 mM citrate buffer (pH 4.8) and combined at a 1:1 ratio (NSCB:MZ) with a final protein concentration of 309 µg/mL and a total cellulase activity of 2 FPU (filter paper unit). The commercial enzymatic cocktail Accellerase 1500 (ACC; Genecor, Rochester, NY, USA) was tested for comparison at a final total cellulase activity of 2 FPU. Saccharification was carried out in 2-mL Eppendorf tubes containing 50 mM citrate buffer (pH 4.5) at 50 °C for up to 48 h in a thermomixer (Eppendorf, Hamburg, Germany) at an agitation rate of 1000 rpm. Samples were collected every 12 h. Each assay was performed in duplicate (biological replicates) and the reducing sugars were measured in triplicate (technical replicates) using the DNS assay [59]. Glucose standards were used to calibrate the glucose over saccharification time. The statistical significance (threshold *p* < 0.05) was determined using Perseus (www.coxdocs.org/doku.php, accessed on 20 July 2021).

## 3. Results

### 3.1. Genomic, Phylogenetic, and Growth Analysis of Staphylotrichum longicolleum FW57

We evaluated the growth of *S. longicolleum* FW57 on six different media, resulting in the formation of pale white to slightly yellow mycelia (Figure 1). Genomic DNA was isolated and analyzed by agarose gel electrophoresis (Appendix A) and the ITS region was amplified and sequenced (Appendix A). Sequencing identified the isolate as a *Humicola* sp. strain, which is preserved at the German Collection of Microorganisms and Cell Cultures (DSMZ) under the identifier DSM105789. The assembled FW57 genome was 35.60 Mbp in length, distributed over six scaffolds with a GC content of 57.51% and an N_50_ scaffold length (weighted median of a contig length needed to cover 50% of the genome) of 5.31 Mbp. The optimal *k*-mer length (subsequences of length *k* contained in the genomic sequence) following assembly with SOAPdenovo was *k* = 15 bp, with a pkdepth (peak depth estimated from *k*-mer distribution) of 29. Gene prediction revealed the presence of 10,979 putative open reading frames (ORFs) with an average of 1665.61 bp per gene or 1489.6 bp per CDS. The genome assembly is available as a biosample from the National Center for Biotechnology Information (NCBI) under the bioproject PRJN413482, accession number JAHCVI000000000, which also contains the number of scaffolds, their sequences, and the annotations. Phylogenetic analysis assigned FW57 with the highest similarity to *Staphylotrichum longicolleum* (formerly known as *Chaetomium longicolleum*, Krzemieniewska & Badura) CBS 100950 (Figure 2).

### 3.2. CAZyme Analysis

The FW57 genomic regions marked as CDS in our de novo assembly were searched for homologs of families (and subfamilies) in the CAZyme database representing enzymes involved in cellulose and sugar metabolism, revealing 596 candidate genes (Figure 3 and Table 1). The candidates were assigned to five different CAZyme classes and associated families (Appendix A).

### 3.3. Evaluation of Enzymatic Activity

We tested the chitinase activity of FW57 on mineral salt medium containing 1% chitin 3 days after inoculation, revealing a value of 0.111 ± 0.003 U/mg (Appendix A). FW57 was cultivated in liquid yeast extract peptone dextrose (YPD) medium, and the enzymatic activity of the supernatant was evaluated. Chitinase activity reached a maximum of ~10.8 ± 0.2 U/mg on day 3 (Appendix A), so we selected this time point to compare the secretome produced under different fermentation conditions.

During SSF (Appendix A), we observed a clear time-dependent difference in the enzymatic activity of the nine CAZyme assays, reflecting the nature of each biomass substrate (Figure 4). When SCB was used, eight of the nine CAZyme activities peaked on day 7, whereas only three activities peaked on day 7 when the substrate was MZ. In most cases, higher activities were observed when the fungus was grown on SCB. The highest enzymatic activities on SCB were xylanases (~30 U/mg on day 7) whereas the highest activity on MZ was a CMCase (~7 U/mg on day 7). Most enzyme activities showed similar time dependencies when cultured on SCB: a peal on day 7 with lower but relatively stable expression at other time points. Exceptionally, laminarinase was not detected on day 7 but was found in the culture supernatant at the other time points with an activity of ~1.3 U/mg. The CAZyme secretome on the MZ substrate was more diverse, with FPases, CMCases, exoglucanases, β-glucosidases and arabinoxylanases showing massive activity on day 7 and subsequently a lower but steady expression similar to that observed on SCB. In contrast, xylanase expression increased from day 7 to 21, laminarinase was expressed similarly to the profile on SCB (not detected on day 7, and 1.5–1 U/mg thereafter), and pectinase and polygalacturanase showed similar trends (only detectable after days 7 and 14, respectively).

### 3.4. Secretome Profiling of S. longicolleum FW57 on Chitinous and Agro-Residual Biomass

We analyzed the FW57 secretome by MS/MS-based proteomics with or without prior fractionation by one-dimensional SDS-PAGE, revealing the presence of 1092 proteins post-fractionation, and 442 in the unfractionated samples (Appendix A). The number of proteins also differed according to the substrate and fermentation type. For liquid fermentation, the numbers ranged from 225 for H-CS to 308 for L-CS, whereas SSF led to the identification of 530 and 807 proteins on MZ and SCB, respectively. We identified 193 proteins solely on chitinous biomass during liquid fermentation, ranging from six on H-CS to 81 on M-CS. We also identified 330 proteins solely in the SCB and MZ secretome fractions, 262 unique to sugarcane, and 68 unique to maize.

Next, we compared the post-fractionation proteins sets co-expressed on each substrate to reveal common profiles (Figure 5A). The largest number of shared proteins was co-expressed when FW57 was grown on the agro-residual biomass (SCB or MZ) with 232 proteins in common, suggesting the fermentation type (SSF) has a strong influence on the secretome. The second largest number of shared proteins was co-expressed when FW57 was grown on all media (110 common proteins), thus representing extracellular housekeeping proteins necessary for growth, including general sugar conversion and homeostasis proteins. Interestingly, the third largest group of shared proteins was common to the chitinous and SCB biomass (28 proteins). The fourth largest was common to the chitinous substrates in liquid fermentation (22 proteins), representing proteins specifically required for this substrate or fermentation type.

Time-course MS analysis was used to identify changes in the *S. longicolleum* FW57 extracellular proteome (in-solution tryptic digestion), during growth on the agro-residual biomass for up to 28 days, with samples analyzed every 7 days. On the MZ substrate, we identified 50, 84, 120 and 135 proteins on days 7, 14, 21, and 28, respectively. FW57 expressed more proteins when cultivated on SCB, with 285, 168, 128 and 156 proteins identified on the same four days (Figure 5B,C). These findings indicate that FW57 can fine-tune the expression of relevant genes to ensure survival in different habitats.

To gain insight into the metabolic diversity of the secretome on different substrates, the identified proteins were classified according to their biological functions (Figure 6A). The annotation is based on the sequences listed in Appendix A. We applied several functional categories, including carbohydrate, energy, lipid, RNA and amino acid metabolism, protein synthesis, redox processes, proteolysis, and proteins with unknown functions. The proteins identified on the chitin and chitosan substrates tended to be distributed similarly according to their molecular functions, with some exceptions. The first difference was found in the distribution of the CAZymes, where an average 17% of the identified proteins on chitosan substrates were CAZymes, but this increased to 22% for H-CS, 21% for SCB and 31% for MZ. Proteins involved in energy metabolism represented ~10% of all identified proteins on the chitosan-like substrates but only to 7% on the agro-residual biomass. Proteins assigned to ‘other biological processes’ constituted 25–29% of the proteins on most substrates but only 20% on MZ. Similarly, proteins involved in redox reactions represented 12–14% on most substrates but only 8–9% for MZ and H-CS. In general, similar expression levels were observed, with the strongest deviations found when FW57 was cultivated on MZ or H-CS.

The substrate-dependent profiles of the 225 CAZymes defend by MS analysis are shown in Figure 6B, and a complete list of the CAZymes identified by in-gel digestion is provided in Appendix A. In general, glycoside hydrolases (GHs) alone and with associated carbohydrate-binding modules (CBMs) were found with the highest relative frequency (56–68%), followed the auxiliary activities (AAs, 20%). The relative frequency of carbohydrate esterases (CEs) was higher than average on the agro-residual biomass, and the frequency of polysaccharide lyases (PLs) was 3–5%, but only in these samples.

For the MS time-course analysis, the number of CAZyme classes detected in the MZ secretome increased with the fermentation time, whereas the number of CAZymes on SCB showed a constant distribution of GHs, AAs, CEs, and PLs over time. This probably reflects the physical, chemical, and morphological characteristics of the steam-exploded lignocellulosic material, given that pretreatment partly hydrolyzes the hemicellulose and reduces the complexity of cell wall components, increasing the purity of cellulose. Accordingly, FW57 has evolved a more robust enzymatic strategy focusing on the degradation of the cell wall (Figure 6C). A list of all CAZymes identified following SSF is provided in Appendix A. The number of enzymes in all CAZyme families increased during cultivation on MZ from 5 to 76, whereas on SCB the largest number of CAZymes was detected on day 7 (91 proteins) and the number then fell to a stable 51–57 for the rest of the fermentation. The relative proportion of GHs (with and without CBMs), AAs and CEs (with and without CBMs) on SCB remained steady at 55–61%, 20% and 14%, respectively. However, on the MZ substrate the proportion of GHs increased from 40% to ~60% and the proportion of AAs declined from 60% to 20% from the first sampling point, whereas the proportion of CEs remained stable at ~13%. The PLs are late CAZymes because their proportion increased from 0–1% on the first sampling day to 4–6% at later time points (Figure 6C).

Some CAZymes were produced on all substrates whereas others were more specific to the molecular and structural composition of a particular type of biomass. The core CAZymes in the secretome included enzymes from every CAZyme class, showcasing the ability of *S. longicolleum* FW57 to break down *N*-acetylglucosamine polymers as well as more complex and recalcitrant substrates. We identified xylanase, amylase, cellulase, esterase, ligninase, pectinase, chitinase, and glycosyltransferase activities, as well as AA families with glucose, galactose, aryl alcohol, gluco-oligosaccharide and chito-oligosaccharide oxidase and oxidoreductase functions (Appendix A).

As anticipated, we detected fewer cellulose, hemicellulose, and pectin degrading CAZymes on the chitinous substrates compared to the agro-residual biomass. The proteins found on all chitinous substrates were GH17 (NEMBOFW57_002419, a predicted β-(1,3)-glucosidase), GH18+CBM18+CBM50 (NEMBOFW57_006926, a predicted chitinase with a modular structure including one CBM18 module and one CBM50 chitopentaose-binding function), and GH32+CBM38 (NEMBOFW57_004148, an invertase with an insulin-binding domain). One CE4 family protein (NEMBOFW57_009578, with predicted acetyl xylan esterase or chitin deacetylase activity) was found on chitin and the agro-residual biomass, but not on the chitosan biomass.

We identified 144 CAZymes in the agro-residual biomass supernatants, including 34 found solely on MZ and 25 solely on SCB. AA families common to MZ and SCB comprised a broad portfolio of enzymes that can modify lignocellulosic material by oxidation (AA families 1, 2, 3, 5, 6, 7, 8, 9 and 16), as well as CEs involved in the degradation of hemicellulose and pectin (families 1, 2, 3, 4, 5, 8, 10, 12, 15, and 16). We identified GH families 1, 2, 3, 5, 6, 7, 10, 11, 12, 15, 16,17, 18, 20, 26, 27, 31, 35, 37, 43, 45, 47, 51, 55, 62, 74, 79, 93, 115, 125, and 131, which hydrolyze all types of saccharides in cell walls, and GH72 with transglycosylation activity. Finally, we identified PL families 1, 3, 4 and 26 (which degrade pectin) on all substrates.

The CAZymes found only on SCB included AA families such as AA1_3 (NEMBOFW57_001206, laccase), AA3_2 (NEMBOFW57_007796, glucose/aryl alcohol oxidase), AA9 (NEMBOFW57_002035, lytic cellulose monooxygenase) and AA12 (NEMBOFW57_008990, pyrroloquinoline quinone (PQQ)-dependent pyranose dehydrogenase). We also identified the CE families CE1 (NEMBOFW57_000278), CE1+CBM1 (FUN 004294 and NEMBOFW57_002052), CE10 (NEMBOFW57_009275 and NEMBOFW57_002792) and CE16+CBM1 (NEMBOFW57_009601), endoglucanases from families GH5 (NEMBOFW57_002806), GH5+CBM1 (NEMBOFW57_004293), GH7 (NEMBOFW57_008456 and NEMBOFW57_003636), GH12+CE1 (NEMBOFW57_007690), GH2 (NEMBOFW57_010681) and GH3 (NEMBOFW57_008747), as well as xylanases from families GH10+CBM1 (NEMBOFW57_003635 and NEMBOFW57_009546), GH15+CBM20 (NEMBOFW57_004103), GH18+CBM1 (NEMBOFW57_006970), GH31 (NEMBOFW57_007820), GH38 (NEMBOFW57_002602), GH45 (NEMBOFW57_009019) and GH67 (NEMBOFW57_004236). Among the eight other proteins solely found on SCB, NEMBOFW57_004511 (GT90+AA9) has an uncommon combination of a glycosyltransferase coupled to a lytic cellulose monooxygenase domain.

The CAZymes identified only on MZ included AA3_2 (NEMBOFW57_009225), AA7 (NEMBOFW57_001464) and two AA9 proteins (NEMBOFW57_001172 and NEMBOFW57_005701), as well as CE1 (NEMBOFW57_009711), CE1+CMB1 (NEMBOFW57_001166) and CE15 (NEMBOFW57_004373), which catalyze the hydrolysis of acetyl groups from hemicellulose (CE1) and pectin (CE15). Many GH families were also identified, catalyzing the cleavage of sugar residues in pectin and/or hemicellulose polymers: the true xylanase GH11 (NEMBOFW57_008340) and promiscuous xylanase GH10+CBM1 (NEMBOFW57_008340), several hemicellulose-debranching families such as GH43 (GH43_5, GH43_6, GH43_21, GH43_24, GH43_24+CBM35, GH43_26+CBM42, GH43_36), GH51 (NEMBOFW57_009294), GH53 (NEMBOFW57_002223), GH54+CBM13+CBM42 (NEMBOFW57_000338), GH62 (NEMBOFW57_004071), GH62+CMB1 (NEMBOFW57_009999), GH75 (NEMBOFW57_006928), GH79 (NEMBOFW57_009514, NEMBOFW57_004002 and NEMBOFW57_008132), GH92 (NEMBOFW57_007118), GH93 (NEMBOFW57_007618), GH95 (NEMBOFW57_006721), GH114 (NEMBOFW57_004027) and GH146+CBM13 (NEMBOFW57_001379). We also detected two PL1 proteins representing subfamilies 1 and 7 (NEMBOFW57_008589 and NEMBOFW57_007080) and one member of PL4 subfamily 3 (NEMBOFW57_006439).

The time-course analysis revealed that at the beginning of SCB degradation (up to 7 days), oxidoreductases, oxidases, cellulases, hemicellulases, and some pectinases were dominant components of the secretome, representing 80.5% of all CAZymes up to 28 days (Appendix A). The abundance of AAs and their isoforms highlights the oxidative capability of *S. longicolleum* during this time, focusing on lignin and cellulose degradation. Among the AA families, we detected seven AA3 alcohol oxidases (NEMBOFW57_001352, NEMBOFW57_008032, NEMBOFW57_010319, NEMBOFW57_010602, NEMBOFW57_008624, NEMBOFW57_010246 and NEMBOFW57_001011), three of which also featured an AA8 cytochrome domain and CBM1. We also detected one AA5 subfamily 1 protein (NEMBOFW57_003104), three AA7 proteins (NEMBOFW57_001310, NEMBOFW57_004115, NEMBOFW57_005776), one AA8 oxidoreductase (NEMBOFW57_008720), six lytic cellulose monooxygenases (NEMBOFW57_001172, NEMBOFW57_004556, NEMBOFW57_005018, NEMBOFW57_007876, NEMBOFW57_001044 and NEMBOFW57_001466, two including CBM1), and one AA12 PQQ-dependent oxidoreductase (NEMBOFW57_003955) that catalyzes the oxidation of sugars with coenzyme PQQ [63]. By day 7, *S. longicolleum* can also disrupt direct ester linkages between carbohydrates and lignin by secreting four CE1 proteins (NEMBOFW57_000278, NEMBOFW57_002054, NEMBOFW57_004513 and NEMBOFW57_007800). Moreover, the repertoire of five reducing-end GH7 proteins on day 7 demonstrates the cooperative mechanism used by oxidative enzymes to create new ends for exoglucanases. Other proteins were only detected on SCB after 14 days: PL1_7 and PL_1_10 (NEMBOFW57_010773 and NEMBOFW57_009693), one AA3 (NEMBOFW57_010319, a predicted alcohol oxidase), and one CE4 (NEMBOFW57_005718, a predicted acetyl xylan esterase or a chitin deacetylase). Additionally, we identified one CE5 cutinase (NEMBOFW57_005577), one CE10 predicted feruloyl/acetylesterase (NEMBOFW57_001299) and 10 GHs with main chain and debranched enzymatic functions for the cleavage of arabinoxylans, β-glucans, arabinogalactans and xyloglucans (GH2+CBM32+CBM51+CBM67, GH5_16, GH7, GH12+CE1, GH15+CBM20, GH18+CBM18+CBM50, GH71, GH79+CE1, GH95, and GH125).

Time-course analysis on the MZ substrate revealed a strikingly different profile. After 7 days, we identified only five CAZymes: AA3 (NEMBOFW57_010319), AA7 (NEMBOFW57_001310), AA9+CBM1 (NEMBOFW57_001044), GH7 (NEMBOFW57_010291) and GH17 (NEMBOFW57_000164). The degradation of MZ cell walls begin properly on day 14, with more diverse CAZyme profiles indicating the potential for oxidoreductase activity, including two more AA3 proteins, one containing an AA8 cytochrome domain (NEMBOFW57_008624), one AA5_1 predicted galactose oxidase, one AA7 predicted oligosaccharide oxidase, and two AA9 proteins (NEMBOFW57_001968 and NEMBOFW57_001466, the latter with a CBM). We also identified families CE4, CE8, CE10, CE12, and CE15 as well as GH2, GH7, GH10, GH43 and GH55, one each of GH6, GH16, GH17, GH18, GH28, GH35, GH79, GH93, GH95 and GH131+CBM1, and PL1_7 and PL4_3. On day 21, we detected further enzymes to facilitate xylan and cellulose degradation: two AA3 proteins (NEMBOFW57_010449 and NEMBOFW57_010602), one CE2 (NEMBOFW57_007038), one GH2+CBM42+CBM67, a β-d-galactofuranosidase modulated with two CBMs from families 42 and 67 specific for arabinofuranose and l-rhamnose (NEMBOFW57_002037), two GH3 proteins with xylan β-(1,4)-xylosidase activity (NEMBOFW57_002874 and NEMBOFW57_002909), three GH7 family glucanases (NEMBOFW57_003080, NEMBOFW57_009077, and NEMBOFW57_010290) as well as GH12 (NEMBOFW57_001184), GH15+CBM20 (NEMBOFW57_007659 a starch-binding glucoamylase), GH16+CBM18 (NEMBOFW57_000084), GH18+CBM18 (NEMBOFW57_001287), GH27 (NEMBOFW57_008762), GH35 (NEMBOFW57_001853), GH37 (NEMBOFW57_003451), GH45 NEMBOFW57_002500), GH54+CBM13+CBM42 (NEMBOFW57_000338), GH62 (NEMBOFW57_001269) and GH115 (NEMBOFW57_005380). In the final sample (day 28), we were still able to detect several oxidative enzymes from families AA3 (NEMBOFW57_008033, NEMBOFW57_009225 and NEMBOFW57_010289) and AA7 (NEMBOFW57_004115), three CEs representing CE4, CE5 and CE16 (NEMBOFW57_005718, NEMBOFW57_007936 and NEMBOFW57_006599), seven GHs mainly related to the breakdown of xylan (GH10), galactan (GH5_16, GH16, GH27), glucan (GH71, GH131) and cellobiose (GH3), and one PL26. Most enzymes detected on MZ were present continuously from their first appearance until the end of the cultivation, but there were six exceptions: GH131+CBM1, a broad-specificity exo-β-(1,3)/(1,6)-glucanase with endo-β-(1,4)-glucanase activity (NEMBOFW57_000683), two GH72 proteins with predicted β-(1,3)-glucanosyltransglycosylase activity (NEMBOFW57_004454 and NEMBOFW57_007083), NEMBOFW57_008762 (a GH27 with α-galactosidase activity), NEMBOFW57_001287 (a GH18+CBM18 chitinase), and NEMBOFW57_000286, a CE12 family protein with acetyl xylan esterase activity.

### 3.5. Conversion of Biomass with the In-House S. longicolleum FW57 Enzyme Mixture

We lyophilized the enzymes secreted on the SCB and MZ substrates after 21 days and resuspended them in 50 mM citrate buffer (pH 4.8) to prepare an in-house mixture (FW57) combining the solutions at a 1:1 ratio (SCB:MZ). The final protein concentration was 309 µg/mL with a total cellulase activity of 2.3 FPU/mL. We then tested the mixture against three different substrates: steam-exploded sugarcane bagasse (XSCB), untreated (in nature) sugarcane bagasse (NSCB) and untreated maize leaves (MZ), each present at a concentration of 5% (*w*/*v*) for 48 h (Figure 7). To compare the performance of our in-house mixture, the same assays were performed with the commercial Accellerase 1500 (ACC) cocktail containing xylanase, exoglucanase, endoglucanase, hemicellulase and β-glucosidase. Both the FW57 mixture and the commercial ACC cocktail were applied with a final cellulase activity of 2 FPU in each reaction (Appendix A).

XSCB showed the most efficient biomass conversion at each time point when using either the in-house FW57 mixture or the commercial ACC cocktail, probably reflecting the ease of access to the substrate following the steam explosion pretreatment, which exposes the cellulose microfibrils [64]. The concentration of glucose released by ACC was 40% higher than FW57 after digestion for 48 h, which is most likely explained by the synergistic activity of the ACC enzymes. After lyophilizing the FW57 secretome, xylo-oligosaccharides, monosaccharides, and products of lignin degradation were also concentrated in the FW57 mixture. These compounds originated from hydrolysis before lyophilization, and were supplemented with the new products of SCB hydrolysis, possibly accumulating to a concentration sufficient to inhibit β-glucosidases, β-xylosidases, cellulases, and xylanases, therefore reducing the efficiency of saccharification [65,66,67]. In the case of MZ, the in-house mixture and commercial cocktail generated similar concentrations of glucose equivalents up to 24 h, but ACC released 40% more reducing sugars after 48 h. The equivalence in released glucose after 24 h demonstrates the significant impact of *S. longicolleum* GHs, forming a complex that was able to progressively access and break down the more soluble components of the MZ hemicellulose matrix, allowing access to the less soluble polymer structures. As observed for SCB, the stagnation of released glucose on MZ after 24 h may also reflect the inhibition of enzymes by excess products from the hydrolysis reactions. The untreated biomass (NSCB) clearly showed the recalcitrance of the lignocellulosic material. The FW57 mixture was able to progressively digest untreated sugarcane bagasse up to 48 h, whereas ACC reached a plateau after 12 h, both treatments leading to the release of similar amounts of glucose. The ACC plateau may also reflect the obstruction of enzymatic access to the cellulose fibers, given that the same phenomenon was not observed on the XSCB substrate. It seems that no plateau is reached for digestion of SCB and NSCB after 48 h applying the FW57 cocktail, so maybe the yield of fermentable sugars could be higher. However, the analysis was focused on the first 48 h, as this would be the critical timeframe for an industrial processes.

## 4. Discussion

*Staphylotrichum longicolleum* FW57 is a fungus isolated from mangrove wood that can digest chitin with remarkable efficiency. Here we assigned this strain to the correct species clade within the family *Chaetomiaceae* and carried out a comprehensive targeted proteomic analysis of its ability to use chitinous biomass and agricultural residues (the C4 crops sugarcane and maize [25,56]). To the best of our knowledge, this is the first comparative analysis of the *S. longicolleum* secretome on different substrates.

Analysis of the 14 proteins produced on all four chitinous substrates revealed only three CAZymes, one of which was a GH18 chitinase with CBMs 18 and 50 (specific for chitin and chito-oligosaccharides [68]) thus appearing especially well-suited for the degradation of chitinous biomass. Among all the *S. longicolleum* CDS revealed by de novo sequencing, we identified 23 GH18 proteins, some combined with CBM1, CBM18, CBM24, or CBM50, but only five different chitinases were present on chitinous biomass. These enzymes most likely break down chitin and chitosan depending on availability, as already shown for the *Serratia marcescens* enzymes SmChiA and SmChiB [69]. SmChiA was ~100-fold less active on chitosan than chitin, but nevertheless degraded both polymers [70]. Importantly, chitinases show varying degrees of promiscuity [71] and deacetylation may not be completed because GH18 needs a correctly positioned *N*-acetyl group at the –1 position in the active site for efficient catalysis [72]. Furthermore, one GH18 (NEMBOFW57_002998) was solely found on chitin biomass, suggesting that this enzyme is dependent on *N*-acetylation. For the other two CAZymes (GH17 and GH32) found on all chitinous biomass, no chitinase activity has been reported thus far. However the GH17 (NEMBOFW57_002419) with a predicted β-(1,3)-glucosidase activity may be involved in fungal cell wall metabolism [73]. The only unique CE found on chitin was CE4 (NEMBOFW57_009578). This enzyme has predicted acetyl xylan esterase (EC 3.1.1.72) or chitin deacetylase (EC 3.5.1.41) activity and would therefore be a good candidate for the enzymatic conversion of chitin to chitosan [74]. Interestingly, the GH75 family chitosanase (NEMBOFW57_006928) was only found on MZ, and no enzymes with potential chitosanase activity from other GH families (GH5, GH8, GH46 and GH80) [75] or any chitin-active lytic polysaccharide monooxygenases (LPMOs) [71] were found on the chitinous biomass. The distribution of CAZymes induced on both forms of agro-residual biomass was more complex, mirroring the complexity of the lignocellulosic matrix. The secretome fractions thus included a lignocellulolytic enzyme mixture with the ability to degrade all cell wall polymers and stored starch granules, including hydrolases, lyases, and oxidative enzymes.

Fungal hydrolytic degradation of cellulose involves at least three steps: (1) internal cellulose bonds are cleaved by endo-β-(1,4)-glucanases (GH5) [76,77,78] to create starting points for cooperative action on shorter polymers; (2) these are digested by exo-β-(1,4)-glucanases and/or cellobiohydrolases (GH7 and GH6) to produce cellobiose, which is (3) converted into two glucose molecules by β-glucosidases (mainly GH1 and GH3, but also some others such as GH39) [79,80]. Enzymes representing all these steps were confirmed in the *S. longicolleum* FW57 secretome. For the first step, a predicted GH5 protein with cellulase activity (NEMBOFW57_009318) was found on both SCB and MZ, whereas two others (NEMBOFW57_002806 and NEMBOFW57_004293) were found only on SCB. For the second step, two GH6 (NEMBOFW57_004785 and NEMBOFW57_008641, predicted endoglucanases or cellobiohydrolases) and eight GH7 proteins were found on SCB, whereas one GH6 (NEMBOFW57_008641) and five GH7 proteins were found on MZ. For the third, six GH3 proteins with predicted β-glucosidase activity were secreted on SCB and one GH1 protein was detected on both substrates.

The degradation of hemicellulose requires enzymes specific for β-(1,4)-linked xyloses, xyloglucan and arabinoxylan acetylated at the C2 and/or C3 positions, as well as β-(1,3), β-(1,4) and β-(1,6) glucan branches [81] that connect pectin to cellulose [17,82]. Phenolic acids that covalently join lignin to arabinoxylan, creating a physical barrier to GHs, must also be removed [83]. The enzymatic portfolio includes endo-β-(1,4)-xylanases (GH10, GH11), α-l-arabinofuranosidases and exo-α-l-(1,5)-arabinanases (GH3, GH43, GH51, GH54, GH62 and GH93), β-xylosidases (GH43 and GH3), acetylxylan esterases (CE1–CE7 CE12), pectin esterases (CE8, CE12, CE15), ferulic acid esterases (CE1) and acetylesterases (CE16) [79]. We identified 10 promiscuous GH10 xylanases and two GH11 proteins that exclusively convert d-xylose substrates [84]. GHs responsible for hemicellulose or rhamnogalacturonan-I (pectin) degradation [85], including two GH27, four GH31, two GH35, two GH93, and two GH115 proteins, cooperated with 12 identified GH43 proteins to convert xylo-oligosaccharides containing arabinose and galactose. Previous secretome analysis in *Trichoderma reesei* and *Aspergillus niger* on SCB [86], *Aspergillus nidulans* on sorghum [82], *Myceliophthora thermophila* on SCB [87], *Nectria haematococca* on maize bran [88], and *Fusarium metavorans* on SCB and MZ [89] identified one, five, eight, four, four, eight, and nine GH43 proteins, respectively. FW57 is thus comparable or even a slightly better producer of GH43 proteins. Additionally, multiple xylan esterases from the CE1, CE2, CE4, CE5, and CE12 families were identified in both secretomes, with 10 CEs on MZ and 11 on SCB, which is comparable to the *T. reesei* [86] and *F. metavorans* [89] secretomes. In contrast, no CEs were found in the secretome fractions of *N. haematococca* on maize bran [88]. We also identified three CE1 feruloyl esterases on MZ and five on SCB. Based on genomic analysis, *S. longicolleum* can express up to six CE1 proteins and five CE1+CBM1 proteins, all but one CE1+CBM1 protein being detected on the agro-residual biomass. The white-rot fungi *Phanerochaete chrysosporium* and *Ceriporiopsis subermispora* have five and three CE1 genes, respectively, but the transcript levels did not increase during the degradation of wood [90,91]. This again demonstrates the potential of *S. longicolleum*, because feruloyl esterases and acetyl xylan esterases increase the accessibility of cellulose to hydrolases [92,93]. Similarly, two CE1 proteins were detected during the growth of *A. nidulans* on sorghum stover [82] and one was detected during the growth of *F. metavorans* on SCB [89].

The GHs we identified represented families GH28 (only found on MZ), GH43 and GH79 (five on MZ, three on SCB) perhaps also including several GH35, GH51 and GH93 proteins (which digest rhamnogalacturonan-I) [94]. We identified one CE8 and one CE12 protein (which remove branches from non-sugar components containing methyl and acetyl groups) on both types of biomass, and three CE15 proteins on each of them. Finally, we identified eight PLs from families PL1, PL3, PL4 and PL26 on MZ, and five on SCB. These cooperate to break down homogalacturonan, rhamnogalacturonan and heteroxylans in our substrates [19] by loosening the cellulose microfibrils to increase accessibility. In contrast, no pectin-digesting GHs, CEs, or PLs were identified in the secretome of *N. haematococca* on maize bran, whereas the *A. niger* BRFM442 secretome contained six GH28, two CE8 and one PL protein on the same substrate [88]. A similar number of CAZymes for pectin degradation was produced by *F. metavorans* on the same biomass [89].

Lignin degradation by fungi involves a complex array of redox enzymes such as oxidases, oxidoreductases, and peroxidases, which produce oxidized saccharides, hydrogen peroxide, and hydroquinones by reducing low-molecular-weight compounds such as oxygen, quinones and metal ions [95]. The *S. longicolleum* secretomes contained 46 AA families among the 117 identified by genome analysis. The families represented with the greatest frequency were AA3 (18 proteins from a total of 26 CDS), AA7 (six proteins from 10 CDS) and AA9 (13 proteins from 33 CDS). The MS data revealed the presence of three cellobiose dehydrogenases (CDHs) from AA3 subfamily 1 (AA3_1), featuring a catalytic flavodehydrogenase domain connected via a flexible linker to an electron transferring cytochrome domain classified as AA8 [95]. However, most of the AA3 proteins were from subfamily 2 (AA3_2), which includes (a) aryl alcohol oxidase/aryl alcohol quinone oxidoreductases that catalyze the oxidative dehydrogenation of several aromatic and aliphatic alcohols while reducing oxygen and quinones to hydrogen peroxide and hydroquinones, and (b) glucose 1-oxidases that oxidize the C1 hydroxyl group in sugars while reducing oxygen to hydrogen peroxide [95,96]. Likewise, AA7 proteins oxidize oligosaccharides while reducing oxygen to hydrogen peroxide [97]. Leveraging this hydrogen peroxide production, AA5 subfamily 1 oxidizes simple aldehydes to carboxylic acids and therefore constitutes one of the central hydrogen peroxide-generating enzymes [95]. This enzyme class was also identified in all the secretomes we analyzed. Hydrogen peroxide and hydroquinones support the class II lignin-modifying peroxidases (AA2) [98,99], and two such proteins (from three CDS) were identified in the secretomes on SCB, MZ, and chitinous substrates.

The oxidative hydrogen peroxide environment promoted by AA3_2, AA7 and AA5_1 activated not only peroxidases but also the LPMOs, classified as family AA9 [100,101]. *S. longicolleum* secreted 13 AA9 proteins, three of which also featured CBM1. In comparison, seven AA9 proteins were identified in *Myceliophtora thermophile* secretomes from SCB and commercial cellulose [87] whereas *Thielavia terrestris* secreted three AA9 proteins from glucose, alfalfa and barley straw [102]. All three fungi secreted an AA9+CBM1 protein [87,102]. LPMOs may act in concert with CDHs, with the electrons generated via CDH acting on cellobiose and then shuttled via heme-binding cytochrome (AA8) to trigger the oxidative mechanism of LPMOs [103,104]. Other enzymes such as AA3_2 and AA7 dehydrogenases can also play an accessory role as electron mediators, supporting the oxidative cleavage of cellulose by LPMOs [105,106]. However, *Trichoderma harzianum* and *T. reesei* possess multiple genes encoding AA9 proteins but none encoding CDHs [105], indicating that alternative electron transfer partners and strategies are required to connect lignin and polysaccharide metabolism [101,107].

One 1,4-benzoquinone reductase (AA6) was present in the MZ and SCB secretomes but not on the chitinous substrates, and given its biological function (degradation of intracellular quinones, hydroquinones and benzaldehyde), this protein may be involved in redox cycling and Fenton’s reaction [108]. This activity is important for the conversion of lignin and cellulose because *S. longicolleum* also detoxifies the highly reactive oxygen environment. We also identified two catalases (Appendix A), providing further evidence of oxidative stress [109] because hydrogen peroxide can inactivate LPMOs (AA9) and glucose oxidases (AA3_2) [97,110]. The presence of one cytochrome heme *b* domain AA8 in the SCB and MZ secretome also supports the electron shuttling mechanism to Fenton’s reaction for non-enzymatic and/or enzymatic cellulose chain cleavage [95]. To complete the lignocellulolytic system, *S. longicolleum* secreted one AA12 protein (PQQ dehydrogenase) that can activate LPMOs due to the presence of CDH-like structural domains [111], and one predicted lytic cellulose monooxygenase (AA16) that is currently uncharacterized.

Our time-course analysis clearly demonstrated that digestion of the plant cell wall is dependent on the accessibility of the biopolymers. The largest number of proteins was detected on the XSCB sample and almost all known biopolymer-degrading enzymes were present on the first sampling day. This high initial activity can be explained by the steam explosion pretreatment, which made the polymers more accessible to the fungus [56]. This contrasts with the MZ sample, where *S. longicolleum* first hydrolyzed the pectins and more soluble β-glucans and arabinoxylans before progressing to the less soluble xyloglucans, xylan, and cellulose matrix. This trend was also observed when analyzing the specific activities in each culture (Figure 4) combined with the MS data for in-solution tryptic digests (Appendix A). All activities decreased over time on the SCB substrate, but most activities increased over time on the MZ substrate, except pectinase. The proteomic data reveal the mixed induction of enzymes involved in hemicellulose and pectin degradation over the whole cultivation period, whereas enzymes for lignocellulose degradation are induced at late time points (21–28 days). This indicates an initial step of selective debranching to further expose the glucan chains, thus increasing cellulose digestibility in a highly redoxactive environment. These results are in line with the time-course analysis of *A. nidulans* on sorghum stover [82] and *A. niger* on SCB [86]. However, *T. reesei* uses SGB via a different strategy, in which fewer debranching enzymes are secreted during the early stages of biomass degradation [86] but the cellulose microfibrils are attacked by enzymes with swollenin-like CBMs to disrupt the cell wall structure without producing glucose [112]. The *S. longicolleum* lignocellulose degradation process therefore appears similar to that deployed by *Aspergillus* strains.

The *S. longicolleum* in-house enzyme mixture apparently hydrolyzes MZ and untreated SCB with the same biomass conversion rate as Accellerase 15000, given that similar amounts of reducing sugars were released (Figure 7). The FW57 in-house mixture may therefore be suitable as an alternative for the commercial cocktail when digesting raw lignocellulosic materials. In this case, the *S. longicolleum* enzyme mixture provided a sustainable and a low-energy process, potentially reducing the final costs of saccharification [113,114,115]. Both Accellerase 15,000 and the in-house mixture achieved better results (conversion to glucose) on the XSCB substrate due to the accessibility of the pre-treated lignocellulosic polymer [56,116]. However, Accellerase 15,000 had a stronger impact on the saccharification of treated sugarcane biomass, increasing the production of reducing sugars by 40% compared to FW57. Finally, the CAZymes in the *S. longicolleum* FW57 secretomes could also be used for the biotransformation of active pharmaceutical ingredients or environmental pollutants, for example through the activity of laccases shown by Frieder Schauer and co-workers [117,118], and for applications in the field of industrial biocatalysis as compiled in several books among others by Peter Grunwald [119].

## 5. Conclusions

The CAZymes identified in this study can be used to enhance the enzymatic saccharification of agro-residual biomass. Our workflow involved strain isolation, genome sequencing, CAZyme analysis, and secretome analysis by mass spectrometry. This revealed 224 relevant enzymes by in-gel digestion and 143 by in-solution digestion. A highly integrated and progressive strategy favored ligninolysis via an oxidative hydrogen peroxide mechanism, based on the sequential action of several proteins (cellobiose dehydrogenases, aryl alcohol oxidases, aryl alcohol quinone oxidoreductases, glucose oxidases, glucose dehydrogenases, glyoxal oxidases, LPMOs, laccases, and class II peroxidases) to modify and degrade cellulose, hemicellulose, and lignin. The hydrolysis of untreated biomass using the in-house *S. longicolleum* enzyme mixture and a commercial enzyme cocktail released similar concentrations of glucose, but the diverse enzyme profiles in the *S. longicolleum* need to be characterized in more detail.

## Figures and Tables

**Figure 1 microorganisms-09-01581-f001:**
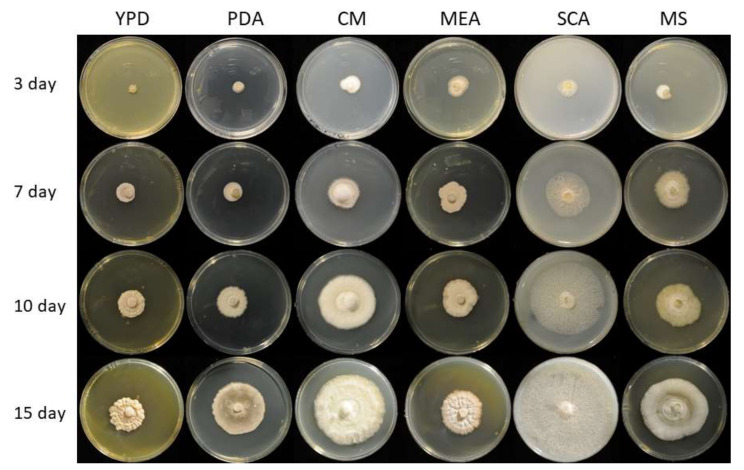
Images of *S. longicolleum* FW57 (DSM105789) mycelia on six different media over 15 days. The selected media were potato dextrose agar (PDA), yeast extract peptone dextrose (YPD), complete medium (CM), malt extract agar (MEA), starch casein agar (SCA) and Mandels’ mineral salts (MS).

**Figure 2 microorganisms-09-01581-f002:**
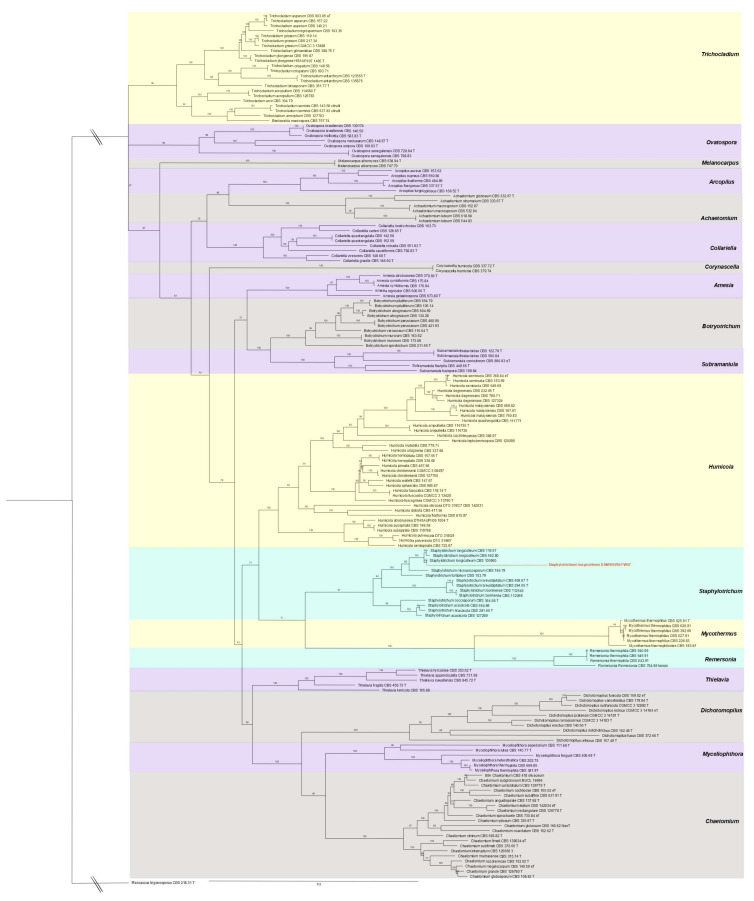
Phylogenetic tree of 173 species of the genus *Chaetomiaceae* plus FW57, estimated by partitioned maximum likelihood bootstrapping. Numbers at internal nodes indicate branch support based on 5000 data pseudo-replicates in IQTree. The tree was rooted at *Microascus trigonosporus* strain CBS 218.31 (redefining *Humicola sensu stricto* and related genera in the *Chaetomiaceae*). The alignment holds 3264 columns and 1802 distinct patterns, of which 1354 are parsimony-informative, 409 are singletons, and 1501 are constant sites.

**Figure 3 microorganisms-09-01581-f003:**
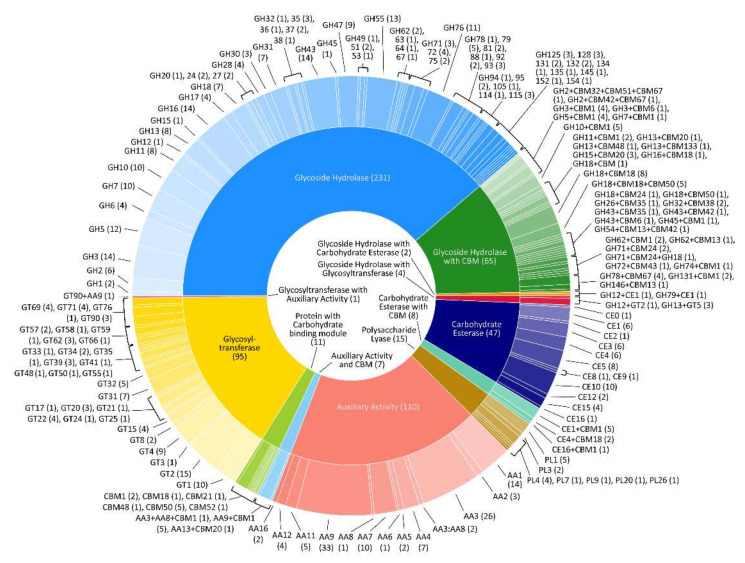
Representation of CAZymes encoded by the *S. longicolleum* genome following the analysis of coding regions revealed by de novo sequencing. The inner ring represents the enzyme classes, and the outer ring names the families. Numbers in brackets represent the frequency of occurrence, also coded by size. Families matching more than one CAZyme category are depicted by strings of activities.

**Figure 4 microorganisms-09-01581-f004:**
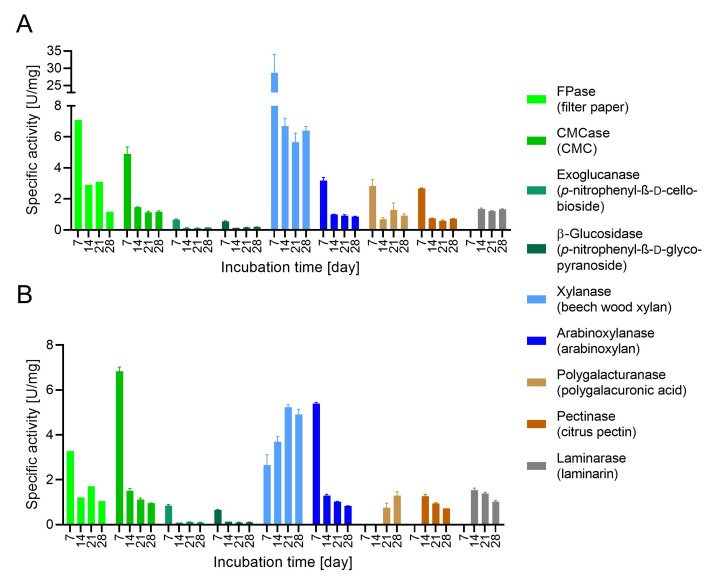
Time-course analysis of the enzymatic activities of FW57 growing on (**A**) sugarcane bagasse (SCB) and (**B**) maize leaves (MZ).

**Figure 5 microorganisms-09-01581-f005:**
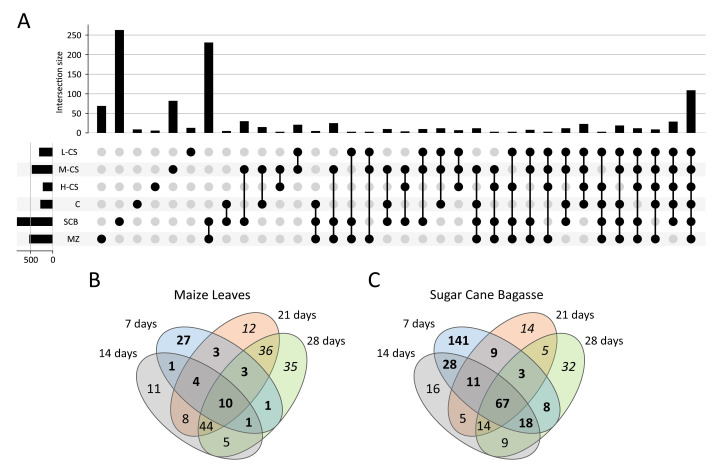
Co-expression of proteins detected by MS-based proteomics. (**A**) In-gel analysis of proteins found in different *S. longicolleum* secretomes. The connected black dots in the lower part of the figure indicate growth conditions resulting in the expression of shared sets of proteins. The number of proteins found under specific conditions is shown by the size of the black bars. The combinations are sorted to first show the unique proteins (specific for certain media) moving toward the set of proteins expressed under all conditions. From all the possible combinations, only those with more than two co-expressed proteins are shown. (**B**,**C**) Venn diagrams showing the FW57 secretome (in-solution analysis) at four time points for two growth conditions on (**B**) maize leaves and (**C**) sugarcane bagasse. Numbers represent the proteins expressed at the indicated time points. Numbers in bold indicate the proteins associated with early growth on the biomass, while those in italics indicate late-stage proteins.

**Figure 6 microorganisms-09-01581-f006:**
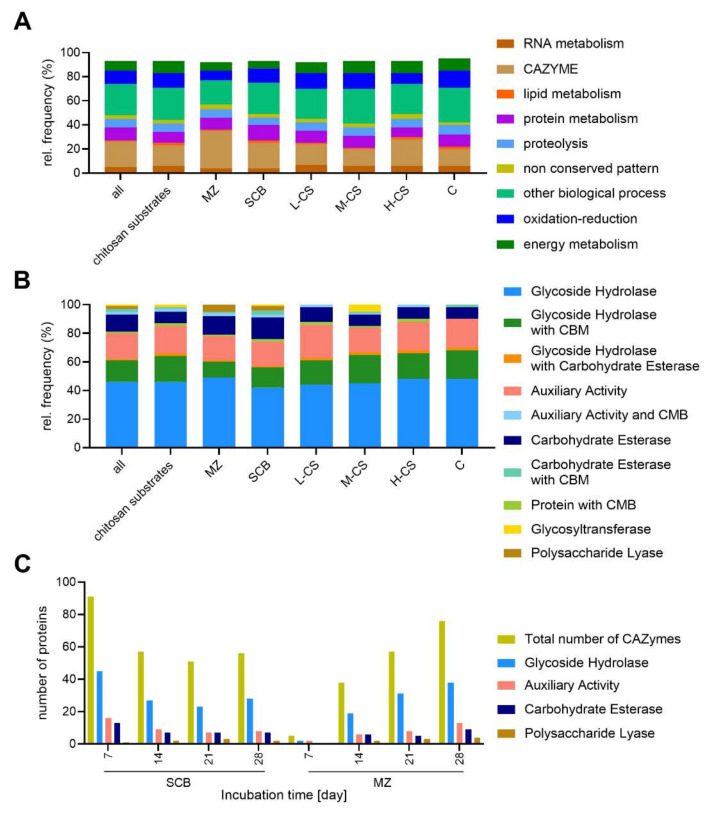
Number of proteins detected by the proteomic analysis of different *S. longicolleum* secretomes. Stacked bar plots are classified according to (**A**) the biological activity of all proteins, or (**B**) the distribution of CAZyme classes. (**C**) Time-course analysis of the CAZyme classes produced by SCB and MZ.

**Figure 7 microorganisms-09-01581-f007:**
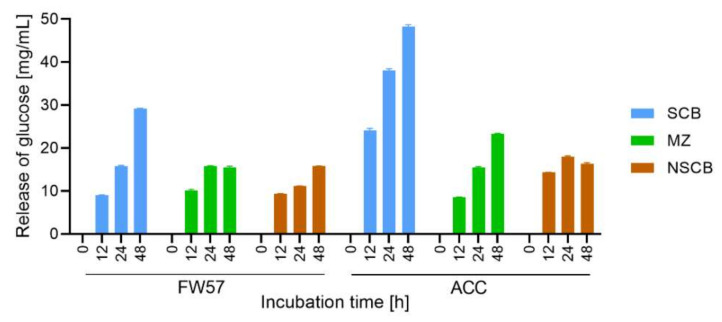
Glucose release by our in-house enzyme mixture and a commercial enzyme cocktail on steam-exploded sugarcane bagasse (XSCB), untreated (in nature) sugarcane bagasse (NSCB) and maize leaves (MZ). The enzymatic mixture was prepared from *S. longicolleum* (FW57), and the commercial cocktail was Accellerase 1500 (ACC).

**Table 1 microorganisms-09-01581-t001:** CAZyme classes identified in the *S. longicolleum* FW57 genome annotation using DIAMOND, Hotpep, and HMMER searched on the dbCAN platform.

CAZyme Classes	Number of Detected Genes
Glycoside hydrolases (GHs)	231
GHs with carbohydrate-binding modules (CBMs)	64
GHs with CBMs with GHs	1
GHs with carbohydrate esterases (CEs)	2
GHs with glycosyl transferases (GTs)	4
CBMs	11
CEs	47
CEs with CBMs	8
Polysaccharide lyases (PLs)	15
PLs with CBMs	0
GTs	95
GTs with CBMs	0
GTs with auxiliary activities (AAs)	1
AAs	108
AAs with AAs	2
AAs with CBMs	6
AAs with AAs with CBMs	1

## Data Availability

All data generated or analyzed during this study are either included in this published article or can be found in the Appendix A.

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
