# Peer review of "Genome and Secretome Analysis of Staphylotrichum longicolleum DSM105789 Cultured on Agro-Residual and Chitinous Biomassâ€"

_microorganisms, 2021, doi:10.3390/microorganisms9081581_

Round 1
Reviewer 1 Report
This work represents a very thorough omics-based dissection of S. longicolleum and with revisions, is suitable for publication in Microorganisms. There is a wealth of data on CAZyme secretion and importantly, activity, which will be useful for researchers in academia and potentially of industrial relevance.
Line 139 – please clarify what is meant by “color formation” – what is the significance of looking for this?
For figure 4, please make connection more clear between CAZyme assay type and enzyme activity measurement (which substrate is used to measure activity of which enzyme?)
Line 666 – please justify why experiments were not continued past 48 hours – if FW57 progressively digests NSCB up to 48h, does it plateau, and if so when?
Line 790-791 – the presence of a putative laccase does not confirm ability to depolymerize lignin. Without experimental evidence of lignin depolymerization (i.e. detection of lignin products), this claim is not substantiated in this manuscript and should be removed.
Will the sequencing data generated be uploaded to a public repository, e.g. mycocosm, NCBI, etc?
Author Response
Dear reviewers,
Your comments were extremely valuable for us to improve our manuscript. We addressed every point as seen in the red answers:
Line 139 – please clarify what is meant by “color formation” – what is the significance of looking for this?
Some ascomycetes, especially Aspergillus sp., are producing different colors depending on the growth media. We wanted to document how the Staphylotrichum longicolleum behaves to the different media.
For figure 4, please make connection more clear between CAZyme assay type and enzyme activity measurement (which substrate is used to measure activity of which enzyme?)
Thank you for this comment. We have clarified this figure by using a unified figure legend that now includes the substrates used to study enzymatic activity.
Line 666 – please justify why experiments were not continued past 48 hours – if FW57 progressively digests NSCB up to 48h, does it plateau, and if so when?
We did not measure digestion of NSCB after 48h as the assay was set up for up to 2 days. Indeed, it seems that no plateau is reached for SCB and NSCB after 48h, but these findings were not crucial in our set up as we tried to mimic industrial processes. Here a pretreatment is done on a relatively short timescale followed by a second fermentation using a different, optimized strain.
Line 790-791 – the presence of a putative laccase does not confirm ability to depolymerize lignin. Without experimental evidence of lignin depolymerization (i.e. detection of lignin products), this claim is not substantiated in this manuscript and should be removed.
The reviewer is correct and as we did not check lignin conversion, we decided to remove that part of the manuscript.
Will the sequencing data generated be uploaded to a public repository, e.g. mycocosm, NCBI, etc?
Yes we have uploaded the assembly to the National Center for Biotechnology Information (NCBI) under the bioproject PRJN413482, accession number JAHCVI000000000 as sqn file, containing the sequence, the scaffolds and the annotation. This information was also included into the manuscript.
Reviewer 2 Report
The manuscript “Genome and secretome analysis of Staphylotrichum longicolleum DSM105789 cultured on agro-residual and chitinous biomass” deals with the isolation, genome sequencing, CAZyme analysis, and secretome analysis by mass spectrometry of a new strain of S. longicolleum. The article is well written, each analysis is well determined and worth to be published in Microorganisms. There is only comment, table 2 and table 3 as they are shown are not clear, considering the number of rows 224 and 134, respectively. I strongly suggest to use another way to visualize the results (e.g. heat-map) or to include them as supplementary materials.
Author Response
Dear reviewers,
Your comments were extremely valuable for us to improve our manuscript. We addressed every point as seen in the red answers:
The manuscript “Genome and secretome analysis of Staphylotrichum longicolleum DSM105789 cultured on agro-residual and chitinous biomass” deals with the isolation, genome sequencing, CAZyme analysis, and secretome analysis by mass spectrometry of a new strain of S. longicolleum. The article is well written, each analysis is well determined and worth to be published in Microorganisms. There is only comment, table 2 and table 3 as they are shown are not clear, considering the number of rows 224 and 134, respectively. I strongly suggest to use another way to visualize the results (e.g. heat-map) or to include them as supplementary materials.
We thank reviewer 2 for his evaluation and we are pleased to read that the manuscript merits publication in Microorganisms. We also agree that the tables 2 and 3 are challenging to read and destroy the flow of the text by their sheer size. We tried to replace the table by a heat-map-like representation, but we ultimately failed. The enzymatic activities are too diverse to generate a concise heat-map and by dividing the tables in several smaller heat-maps the whole section becomes extremely complex and hard to decipher. We therefore reorganized the manuscript and put table 2 and 3 in the SI. In this way, we were able to restore the flow of the text and maintained the machine readability of the tables, which ultimately will be relevant for readers of Microorganisms.